# Novel Lung Growth Strategy with Biological Therapy Targeting Airway Remodeling in Childhood Bronchial Asthma

**DOI:** 10.3390/children9081253

**Published:** 2022-08-19

**Authors:** Mitsuru Tsuge, Masanori Ikeda, Hirokazu Tsukahara

**Affiliations:** 1Department of Pediatric Acute Diseases, Okayama University Academic Field of Medicine, Dentistry, and Pharmaceutical Sciences, Okayama 700-8558, Japan; 2Okayama University School of Medicine, Okayama 700-8558, Japan; 3Department of Pediatrics, Okayama University Academic Field of Medicine, Dentistry, and Pharmaceutical Sciences, Okayama 700-8558, Japan

**Keywords:** bronchial asthma, chronic obstructive pulmonary disease, lung function trajectory, type 2 inflammation, airway remodeling, omalizumab, mepolizumab, benralizumab, dupilumab

## Abstract

Anti-inflammatory therapy, centered on inhaled steroids, suppresses airway inflammation in asthma, reduces asthma mortality and hospitalization rates, and achieves clinical remission in many pediatric patients. However, the spontaneous remission rate of childhood asthma in adulthood is not high, and airway inflammation and airway remodeling persist after remission of asthma symptoms. Childhood asthma impairs normal lung maturation, interferes with peak lung function in adolescence, reduces lung function in adulthood, and increases the risk of developing chronic obstructive pulmonary disease (COPD). Early suppression of airway inflammation in childhood and prevention of asthma exacerbations may improve lung maturation, leading to good lung function and prevention of adult COPD. Biological drugs that target T-helper 2 (Th2) cytokines are used in patients with severe pediatric asthma to reduce exacerbations and airway inflammation and improve respiratory function. They may also suppress airway remodeling in childhood and prevent respiratory deterioration in adulthood, reducing the risk of COPD and improving long-term prognosis. No studies have demonstrated a suppressive effect on airway remodeling in childhood severe asthma, and further clinical trials using airway imaging analysis are needed to ascertain the inhibitory effect of biological drugs on airway remodeling in severe childhood asthma. In this review, we describe the natural prognosis of lung function in childhood asthma and the risk of developing adult COPD, the pathophysiology of allergic airway inflammation and airway remodeling via Th2 cytokines, and the inhibitory effect of biological drugs on airway remodeling in childhood asthma.

## 1. Introduction

### 1.1. Childhood Asthma

Bronchial asthma is a chronic respiratory disease characterized by airway obstruction and hyperresponsiveness [1], with inflammation of the airways, resulting from exposure to allergens or other environmental irritants, causing bronchoconstriction, wheezing, and shortness of breath. The global prevalence of childhood asthma is estimated to be 5–10%, with significant differences between countries [2]. The prevalence of childhood asthma is reported to be 8.4% in the United States [3] and 9.4% in the European Union [4].

Although the severity of asthma varies, most childhood asthma is controlled using low-to-medium doses of inhaled corticosteroids (ICS), with or without additional controller drugs. However, despite treatment with high doses of ICS, and long acting β2 agonists or oral corticosteroids, some patients with severe asthma exhibit persistent asthma symptoms, frequent deterioration, and decreased lung function. Uncontrollable persistent asthma occurs in 5–7% of children with asthma, with geographical differences, and approximately 1 in 100–150 children has severe asthma [5,6,7]. Asthma in childhood causes allergic inflammation in the airways and airway remodeling, leading to persistent airflow obstruction and decreased respiratory function [8]. Children with persistent asthma exhibit a high frequency of asthma attacks, pulmonary dysfunction, poor quality of life, risk of medication-related side effects, and constitute a high medical burden for treatment and emergency consultations. A computerized search was performed using PubMed, combining the terms (asthma OR COPD) AND (trajectory OR lung development OR lung growth) AND (baby OR child OR pediatrics) with an English language filter, to identify studies on the lung growth trajectory in children with asthma, published until 30 June 2022. Furthermore, references within the included articles were scanned for other relevant papers. We excluded articles on lung growth in other chronic lung diseases were excluded.

### 1.2. Natural Prognosis of Lung Function in Childhood Asthma

Approximately 80% of children with bronchial asthma develop this condition by the age of 3 years, but most of these patients remit after puberty and exhibit remission of respiratory symptoms [9]. Hence, bronchial asthma is considered to be a disease with a good prognosis [10]. However, a large-scale study has reported a low proportion of children with asthma who achieve “complete remission” without asthma symptoms or inhaled corticosteroid use, normal lung function, and no bronchial hyperreactivity [11], suggesting that asthma severity at a young age is directly related to that in adolescence and adulthood. In a study of patients with moderate-to-severe pediatric asthma, 15% were in remission, 22% had intermittent asthma, and the rest had persistent asthma [12]. In a cohort study in Melbourne, Australia, approximately half of the 7-year-old patients with pediatric asthma were not in remission at 50 years of age, and only 15% of 7-year-old patients with severe asthma were in remission at 50 years of age [13]. Many cohort studies investigating lung function in childhood asthma from childhood to adolescence have reported that persistent wheezing was associated with decreased lung function development in adolescence [14,15,16]. In one cohort study, lung function was followed in children between the ages of 9 and 26 and 25% of the children had persistent or recurrent wheezing from childhood, with consistently low lung function being observed in these patients [17]. In the Childhood Asthma Management Program (CAMP) study, approximately one-third of participants presented with decreased lung function, and 5–10% of children with asthma experienced severe asthma in adulthood [18]. These studies indicate that asthma that develops in childhood may persist into adulthood, and patients with persistent asthma in adulthood already have decreased lung function from childhood. Furthermore, childhood asthma is strongly associated with respiratory depression and long-term prognosis in adulthood, and some patients develop COPD with irreversible airflow limitation [19].

### 1.3. Decreased Respiratory Function in Childhood and Onset of COPD in Adulthood

COPD is a disease with a high global incidence and mortality. COPD is caused by abnormalities in the airways and alveoli, characterized by persistent respiratory symptoms and irreversible airflow limitation [20]. COPD typically presents with small airway obstruction in the early stages and progresses slowly with diminished lung function. The onset of COPD is affected by genetic and environmental factors, with exposure to harmful particles and gases (particularly tobacco) constituting the main risk factor [21]. However, smoking is not the sole factor in the development of COPD, as only 10% of smokers develop COPD and 25–50% of COPD patients are non-smokers [22,23].

Patients with COPD already have diminished lung function in childhood, and the onset of COPD in adulthood may result from diminished peak respiratory function that emerges in adolescence [24]. Low forced expiratory volume in one second (FEV1) levels in adolescence increase COPD morbidity and mortality [25], and decreased lung function at 7 years of age is a risk factor for developing both asthma and COPD in adulthood [26]. A study that followed the trajectory of lung function from childhood to adulthood reported that children with persistently low lung function had severe and recurrent wheezing associated with exposure to tobacco smoke [27]. In one pediatric cohort study, patients with rapid decline in lung function in adulthood and those with low lung function in adolescence were more likely to develop COPD compared to participants with a normal rate of decline in lung function [28]. These studies indicate that the onset of COPD involves smoking in adulthood as well as decreased lung function in childhood [29,30,31] (Figure 1).

### 1.4. Lung Development in Childhood

Lung development is a continuous lifelong process that begins in the womb and continues from birth to adolescence and early adulthood, and is referred to as the lung function trajectory. The lungs develop by 20 years of age, resulting in increased lung size and number of alveoli and structural complexity of the lung [32,33]. Lung function plateaus between 20 and 25 years and then gradually declines with age. There are sex differences in lung development, with maximal lung function achieved in women at 15–20 years of age and in men at 20–25 years of age [34]. Lung growth in childhood is critical for achieving peak lung function in adulthood.

However, lung development in childhood is affected by various factors, including genetic predisposition, maternal smoking, preterm birth, neonatal chronic lung disease, passive smoking, childhood asthma, childhood respiratory infections, poor nutrition, and air pollution [19,31,35,36]. Impairment in lung function development due to these factors can lead to inadequate peak lung function in adolescence and predispose to COPD in adulthood. Case-control studies of former adult smokers and current smokers reported that a history of childhood pneumonia and/or asthma was associated with the onset or exacerbation of COPD [37]. A large study reported that risk factors for COPD included asthma in parents, exposure to smoking in the prenatal period and childhood, preterm birth, low birth weight, lower respiratory tract infections, recurrent asthma, childhood asthma, and second-hand smoking [38]. Reports indicate that childhood wheezing and decreased lung function due to parental smoking are associated with a high risk of developing COPD in adulthood [8,39].

### 1.5. Impaired Lung Growth, and Increased Risk of Developing COPD, Due to Childhood Asthma

Childhood asthma is a cause of impaired lung growth in childhood, and is one of the risk factors for COPD, similar to smoking [30,40,41]. In a retrospective study in adults, 19% of patients with COPD had a history of childhood asthma, and the risk associated with childhood asthma was equivalent to smoking 62 packs year [42]. A study reported that adults aged 50 years with a history of severe asthma between the ages of 6 and 7 years had a 32-fold higher risk of COPD compared to those without such a history [30]. In addition, a study of adult patients with asthma and COPD identified airflow obstruction and decreased lung volume in adolescence as strongly involved in the onset of COPD in adulthood [43]. A prospective cohort study of children aged 10–15 years reported that childhood asthma was associated with an increased risk of COPD [40], and another pediatric cohort study showed that children with childhood asthma were more likely to be treated or hospitalized with COPD during adulthood [44]. In the CAMP cohort study, 11% of patients with pediatric asthma met COPD criteria based on spirometry in adolescence, and more patients with a low growth pattern of lung function developed COPD compared to those with a normal growth pattern [45]. Various longitudinal studies on long-term lung function trajectories for childhood asthma have reported that childhood asthma impairs lung function maturation, and subsequent decline in lung function in adolescence and adulthood can lead to COPD [17,30,46,47,48,49,50,51]. In this regard, early treatment strategies tailored to the severity of childhood asthma may improve pulmonary dysfunction and reduce the risk of developing COPD in the future.

## 2. Airway Inflammation and Remodeling with T2-High Asthma in Children

### 2.1. T2-High Asthma in Childhood Asthma

Two major end types of bronchial asthma are currently recognized, based on the type of underlying airway immune inflammation, defined as type 2 (T2)-high asthma and T2-low asthma [52]. T2-high asthma encompasses more than 50% of the endotypes of asthma, and most patients with pediatric asthma have T2-high asthma. T2-low asthma is more common in adults than in children. T2-high asthma includes allergic non-eosinophil, non-allergic eosinophil, and mixed granulocytic phenotypes. It is characterized by increased peripheral blood eosinophil counts, exhaled nitric oxide fractions, and allergen-specific immunoglobulin E (IgE) levels [53].

T2-high asthma involves T helper 2 (Th2) cells, eosinophils/basophils, and group 2 innate lymphoid cells (ILC2), in conjunction with epithelial cell-derived cytokines, such as interleukin (IL)-25, IL-33, and thymic stromal lymphopoietin (TSLP), leading to allergic sensitization and airway inflammation sustained by IgE, IL-4, IL-5, and IL-13 [54]. IgE is mediated by Th2 cells and produced by plasma cells via the release of IL-4 and IL-13 cytokines. IgE binds to the high-affinity IgE receptor (FcεR1) expressed on mast cells, basophils, dendritic cells, airway smooth muscle cells, and eosinophils. Cross-linking with allergens releases pro-inflammatory mediators, such as histamine, prostaglandin D2, leukotriene C4, tryptase, and chimase, causing bronchoconstriction and airway inflammation [55,56]. IL-5 is produced by mast cells, Th2 cells, and ILC2 cells and binds to the IL-5 receptor (IL-5R). IL-5 mobilizes eosinophils from the bone marrow and is involved in eosinophil differentiation, proliferation, and activation in the respiratory tract. Induction of eosinophil inflammation promotes airway remodeling, subepithelial thickening, goblet cell metaplasia, and changes in mucin composition in sputum [57]. IL-4 and IL-13 are produced by mast cells, Th2 cells, and ILC-2 cells. These cytokines bind to the type 2 receptor complex (IL-4Rα/IL-13Rα1) on airway epithelium, smooth muscle cells, eosinophils, and mast cells [58]. IL-4 induces T cell differentiation into Th2 lymphocytes, induces B cell class switching, and upregulates IgE synthesis [59]. IL-13 directly affects airway contraction, increases airway hypersensitivity and mucus production, stimulates periostin release from airway epithelial cells, and plays a key role in airway remodeling [60] (Figure 2).

### 2.2. Th2 Cytokines and Airway Remodeling

In a subset of patients with asthma, variable airway restriction progresses to persistent airway restriction or fixed airway obstruction, leading to persistent asthma symptoms, worsening of lung function, airway dilation, and poor response to bronchodilators [30]. Airway restriction in asthma is associated with bronchoconstriction, airway edema, mucus secretion, airway hyperresponsiveness, as well as irreversible pathological structural changes in the airways, called airway remodeling [61,62]. Airway remodeling tends to worsen in parallel with increasing asthma severity, but it is also observed in patients with mild asthma [63]. In addition, airway remodeling has been observed in preschoolers and school-aged patients with asthma, indicating that airway remodeling may have already occurred early during asthma onset [64,65]. Indeed, exposure to asthma triggers results in a rapid increase in airway remodeling parameters [66].

Pathological changes in airway remodeling include loss of airway epithelial cells, goblet cell hyperplasia, myofibroblast proliferation, airway smooth muscle cell hyperplasia, submembrane tissue fibrosis, and increased angiogenesis [67]. One cause of pathological changes in the airways is Th2 inflammation caused by crosstalk between different cell types in the airway wall and submucosa [68]. Th2 inflammation-related cytokines, such as IgE, IL-4, IL-5, and TSLP, can release chemokines and pro-inflammatory and/or fibrosis-promoting cytokines [69,70,71,72,73]. Downstream factors of these Th2 inflammatory mediators include platelet-derived growth factor, transforming growth factor β (TGF-β), and fibroblast growth factor, which promote the production of extracellular matrix (ECM) proteins that contribute to the pathogenesis of airway remodeling [74,75]. ECM proteins promote the proliferation of airway smooth muscle cells and remodeling in vivo [76]. In particular, TGF-β is considered to be an important factor in the airway remodeling of asthma. Subepithelial fibrosis is a major factor in airway remodeling, and TGF-β1 production by immunocompetent cells in asthmatics induces epithelial-mesenchymal transformation (EMT), which transforms airway epithelial cells into mesenchymal cells [74,77,78,79]. In addition, TGF-β is activated in response to the mechanical environment of the airways, such as bronchoconstriction and increased airway wall rigidity, leading to EMT, ECM protein production by fibroblasts, and airway smooth muscle cell proliferation [80] (Figure 3).

## 3. Efficacy of Biological Drugs in Childhood Asthma and Effects on Airway Remodeling

The underlying endotype of the pathology of childhood asthma was identified as Th2-high asthma by surrogate markers of airway T2 inflammation. As a result, biological drugs targeting T2 inflammatory mediators have been developed, enabling personalized treatment strategies for severe childhood asthma [81,82,83]. Long-term uncontrolled airway inflammation due to childhood asthma promotes airway remodeling, leading to serious and long-term impairment of lung function [62]. Considering the immunomodulatory effects of these biologics, inhibitory effects may be exerted on airway remodeling. Currently, the biologics approved for the treatment of severe asthma in children are omalizumab, mepolizumab, benralizumab, and dupilumab. The use of these biologics may suppress asthma symptoms and improve the natural course of childhood asthma by suppressing airway remodeling.

### 3.1. Omalizumab

Omalizumab is a humanized anti-IgE monoclonal antibody that binds to the Fcε3 segment, which is the constant region of free IgE. It prevents free IgE binding to IgE high affinity receptors (FcεR1) receptors on the surface of mast cells, basophils, and dendritic cells, and reduces the downstream allergic inflammatory cascade [84]. It also reduces free IgE circulating in the reticular endothelial system, resulting in a further reduction in allergic reactions [85]. In addition, omalizumab reduces the expression of FcεRI in basophils, mast cells, and dendritic cells, and suppresses the release of Th2 cytokines [86]. Omalizumab was approved by the U.S. Food and Drug Administration (FDA) and European Medicines Agency (EMA) in 2003 as an additional treatment for moderate-to-severe allergic asthma in children and adults aged 6 years and older [87,88].

In a randomized controlled trial of children aged 6–12 years with moderate-to-severe allergic asthma, the omalizumab group exhibited decreased asthma exacerbations, hospitalization rates, and doses of oral corticosteroids; and improved lung function and quality of life [89,90,91]. Omalizumab has also been reported to suppress asthma attacks in spring and autumn. A multicenter study of children and adolescents, aged 6–20 years, with allergic persistent asthma reported that asthma attacks in spring and autumn were reduced in the omalizumab group relative to the placebo group [92]. A study of 478 pediatric patients with severe asthma demonstrated that omalizumab reduced the rate of asthma attacks in the fall and enhanced the peripheral blood mononuclear cell IFN-α response to rhinovirus in vitro [93,94].

### 3.2. Mepolizumab

Mepolizumab is a mouse humanized immunoglobulin G1 monoclonal antibody that selectively binds to circulating IL-5 and prevents the binding of IL-5 to IL-5R on eosinophils and reduces eosinophil activation [95]. It is currently approved by the FDA and EMA as an additional maintenance therapy option in patients with severe eosinophilic asthma over 6 years of age [96]. Clinical trials investigated the efficacy of mepolizumab in patients with eosinophilic asthma aged 12 years and older and reported reduced asthma exacerbations, emergency visits, and hospitalization rates, as well as improvements in FEV1 and quality of life [97,98]. A 52-week open-label extension study reported a long-term improvement in asthma exacerbations and reduction in oral corticosteroid doses [99]. A long-term extended safety and efficacy study in patients with asthma reported a decrease in the asthma exacerbation rate and blood eosinophil counts alongside improved asthma control [100]. In a European multicenter study, children aged 6–11 years with severe eosinophilic asthma were treated with mepolizumab for 52 weeks, reducing decreased blood eosinophil counts and the frequency of asthma attacks [101,102]. Various studies have reported on the efficacy rates of this treatment for eosinophilic asthma [103,104].

### 3.3. Benralizumab

In contrast to mepolizumab, which binds to IL-5 itself, benralizumab is a humanized monoclonal antibody that binds to the α-subunit of IL-5R (IL-5Rα) on the surface of eosinophils and basophils. Benralizumab blocks the binding of IL-5 to its receptors and inhibits eosinophil differentiation and maturation in the bone marrow. In addition, this antibody enhances binding to the FcγRIIIA receptor on natural killer cells, accelerating apoptosis of circulating and tissue-existing eosinophils through antibody-dependent cellular cytotoxicity [105]. In a large multicenter phase III study in patients including adults and adolescents, benralizumab treatment in patients with severe eosinophilic asthma resulted in improvements in asthma symptoms, respiratory function, and quality of life, as well as a reduction in annual asthma exacerbations and doses of oral steroids [106,107,108]. Benralizumab was approved by the FDA and EMA in 2017 as additional maintenance therapy for patients with severe eosinophilic asthma over the age of 12 years. Benralizumab is approved for use in children over 12 years of age in the United States, but is only approved for use in adults in Europe. Further research on the efficacy and safety of benralizumab in the pediatric population under 12 years is warranted.

### 3.4. Dupilumab

Dupilumab is a humanized monoclonal antibody that targets the IL-4 receptor alpha chain (IL-4Rα) and inhibits signal transduction mediated by IL-4 and IL-13 by binding to the receptor [109]. Dupilumab was approved by the FDA and EMA as additional maintenance therapy for patients with moderate-to-severe asthma over the age of 12 years with elevated blood eosinophils and/or fractional exhaled nitric oxide (FeNO) [110].

A study of patients aged 12 years and older, with moderate-to-severe asthma, investigated the efficacy of dupilumab. Administration of dupilumab decreased the rate of exacerbation of asthma and improved lung function compared with placebo [111,112]. Greater efficacy was observed in patients with high baseline blood eosinophil counts and high FeNO. Post-hoc analysis of subgroups of adolescents aged 12–17 years revealed improved lung function and reduced asthma exacerbations [113]. In a study of glucocorticoid-dependent patients with severe asthma over 12 years of age, a 24-week administration of dupilumab reduced oral glucocorticoid use, reduced asthma exacerbations, and improved lung function, compared to the placebo group [114]. In addition, in a study of children aged 6 to 12 years with persistent asthma, additional doses of dupilumab reduced asthma exacerbations and improved lung function and asthma control compared to placebo [115]. Further research on the efficacy and safety of dupilumab in the pediatric population under 12 years is warranted.

## 4. Inhibitory Effect of Airway Remodeling by Biological Drugs

The most reliable way to assess structural changes in airway remodeling is pathological examination of biopsied bronchial or lung tissue [116]. Intrabronchial biopsy using a bronchoscope is performed for the purpose of differential diagnosis or evaluation of the effect of asthma treatment. Although it is possible to determine the thickening of submucosal tissue and changes in smooth muscle mass mainly in the proximal central airway, it is difficult to collect tissue in the peripheral small airways. Transbronchial biopsy is possible to collect peripheral airway tissue, but it is highly invasive. Both tests are difficult to perform, especially in pediatric patients, and can cause bleeding and pneumothorax complications.

As a non-invasive examination to evaluate the airway structure, an evaluation utilizing recent advances in image analysis technology has been reported [117,118,119]. In particular, airway morphology analysis using computed tomography (CT) can objectively quantify the degree of structural change in the bronchi [120,121,122]. Changes in airway structure obtained by this analysis correlate with airflow restriction and pathological change [123,124,125]. Furthermore, analysis using high-resolution CT enables analysis of bronchial trees, structural analysis of the distal airway, and area measurement of the airway wall cross section, making it possible to evaluate the degree of airway remodeling more accurately [126,127,128,129,130]. However, CT airway analysis using three-dimensional (3D) images has been used mainly in adult studies to date [119,129,131], and 3D-CT airway analysis was considered to be difficult in children due to their small size. A previous study using two-dimensional CT image analysis has been reported in pediatric asthma [132], but recent advances in CT equipment and image analysis software are enabling CT analysis of airway structures in children [133]. However, the airway structure obtained by CT includes not only irreversible tissue changes, but also reversible changes in the airway mucosa. In addition, there is a risk of radiation exposure, and there are restrictions in terms of practicality and ethics.

Biological agents currently used for childhood asthma targeting Th2-high asthma may affect airway remodeling, and CT airway analysis, as well as pathological assessment of biopsied airway mucosa, is revealing their effects (Table 1). Three studies using biopsied bronchial tissue before and after omalizumab treatment in adult patients with severe asthma showed a significant reduction in bronchial reticular basement membrane thickness, airway smooth muscles, and fibronectin deposition [134,135,136]. Furthermore, three studies using CT airway analysis gave similar assessments, reporting that omalizumab treatment in adult patients with severe asthma significantly reduced airway wall thickness and area [137,138,139]. The improving effect on airway remodeling has been confirmed not only with omalizumab, but also with other biological drugs. CT airway analysis before and after administration of mepolizumab for severe asthma in adults showed a significant decrease in the airway wall area after treatment [140]. It was also reported that benralizumab administration reduced airway smooth muscle mass by 29% in adults with severe asthma [141]. There are no reports of animal model or human studies on the effect of dupilumab on improving airway remodeling. Unfortunately, all the above studies have been conducted in adults with severe asthma, and no studies have demonstrated a suppressive effect on airway remodeling in childhood severe asthma. An ongoing Italian clinical trial is conducting CT airway analysis using a deep learning model prospectively in children with severe asthma to develop a prognostic scoring (ClinicalTrials.gov number, NCT05140889). Further investigation is needed to determine whether biological treatment from childhood prevents the progression of airway remodeling.

## 5. Conclusions

In childhood asthma, chronic airway inflammation impairs normal lung growth, even after remission of respiratory symptoms. The inability to obtain peak lung function in adolescence affects lung function in adulthood, which leads to a risk of developing COPD. Therefore, early suppression of airway inflammation in childhood asthma and prevention of asthma exacerbations may improve lung maturation, leading to good lung function and prevention of COPD in adulthood. As airway inflammation due to T2-high asthma in childhood causes persistent pathological airway remodeling with asthma exacerbations, administration of biologics targeting Th2 cytokines may suppress the progression of airway remodeling and prevent pulmonary dysfunction in adulthood. It is necessary to conduct clinical trials using minimally invasive evaluation methods, such as CT airway analysis, to ascertain the inhibitory effect of biological drugs on airway remodeling in severe childhood asthma.

## Figures and Tables

**Figure 1 children-09-01253-f001:**
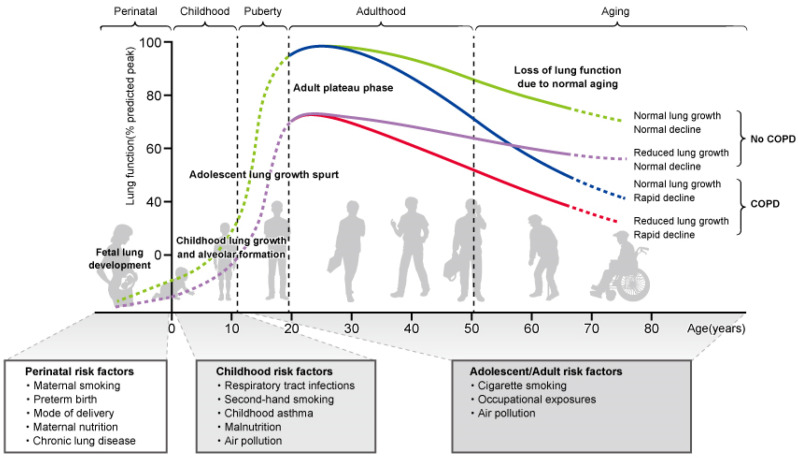
Patterns of growth and decline in lung function and risk factors leading to COPD.

**Figure 2 children-09-01253-f002:**
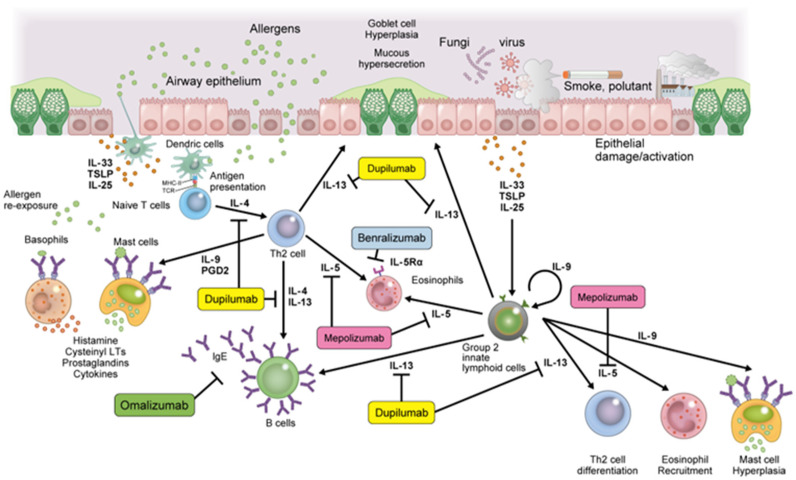
Th2 allergen responses in the asthmatic airway and suppression effect of biological drugs in childhood asthma.

**Figure 3 children-09-01253-f003:**
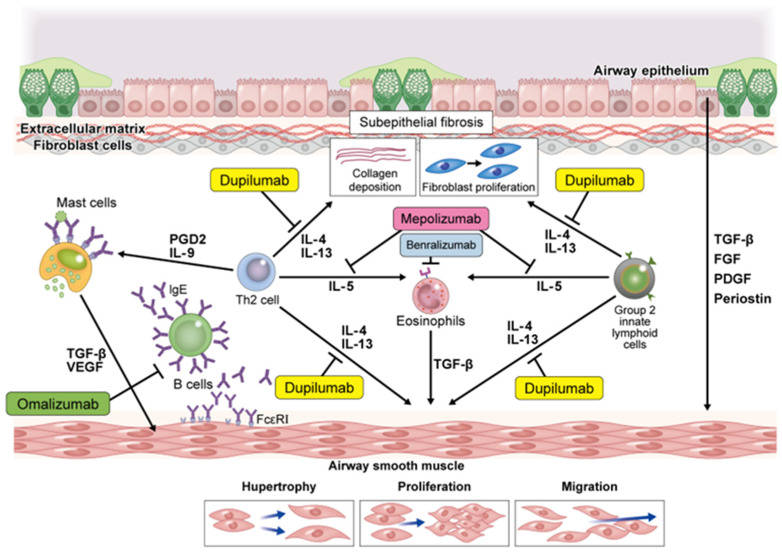
Induction of Th2 cytokines for airway remodeling and suppression by biological drugs.

**Table 1 children-09-01253-t001:** Summary of biological drugs in vivo effects on airway remodeling.

Drug	Year	Analysis	Study	Ref.
Omalizumab	2012	Bronchial biopsies	11 adult severely allergic asthmatics, 12-month treatment - Reduction in reticular basement membrane thickness	[134]
Omalizumab	2012	CT airway analysis	14 adult severely allergic asthmatics, 16-week treatment - Reduction in airway wall thickness and wall area - Increase in the tracheal lumen area	[137]
Omalizumab	2014	CT airway analysis	26 adult severely allergic asthmatics, 48-week treatment - Reduction in airway wall thickness	[138]
Omalizumab	2017	Bronchial biopsies	8 adult severely allergic asthmatics, 36-month treatment - Reduction in reticular basement membrane thickness - Reduction in airway smooth muscle proteins	[135]
Omalizumab	2018	CT airway analysis	12 adult severely allergic asthmatics, >4-month treatment - Reduction in airway wall area	[139]
Omalizumab	2020	Bronchial biopsies	13 adult severely allergic asthmatics, >12-month treatment - Reduction in reticular basement membrane thickness - Reduction in fibronectin deposit in airway submucosa	[136]
Mepolizumab	2009	CT airway analysis	29 adult severely allergic asthmatics, 12-month treatment - Reduction in airway wall area	[140]
Benralizumab	2019	Bronchial biopsies	15 adult severely allergic asthmatics, 12-month treatment - Reduction in airway smooth muscle proteins	[141]
Dupilumab			No study on in vivo effects on airway remodeling	

## Data Availability

Not applicable.

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
