# Peer review of "Novel Lung Growth Strategy with Biological Therapy Targeting Airway Remodeling in Childhood Bronchial Asthma"

_children, 2022, doi:10.3390/children9081253_

Round 1

Reviewer 1 Report

The paper "Novel lung growth strategy with biological therapy targeting airway remodeling in childhood bronchial asthma" by Mitsuru Tsug et al. is a solid analysis of the literature related to this topic. There some minor issues that can be solved. 

1. Abstract. You can add a short sentences with the idea from the page 8, line 329: "no studies have demonstrated a suppressive effect on airway remodeling in childhood severe asthma"

2. Page 2, line 54. Can you define the concept of the "spontaneous remission rate in childhood asthma?"

3. Page 3, line 123. Packs year instead of " packs/year". 

4. Page 8, line 309. "Although CT airway analysis has been used mainly in adult studies to date [119,129,131,132]"-   reference 132 is related to children : Marchac, V.; Emond, S.; Mamou-Mani, T.; Le Bihan-Benjamin, C.; Le Bourgeois, M.; De Blic, J.; Scheinmann, P.; Brunelle, F. Thoracic CT in pediatric patients with difficult-to-treat asthma. AJR Am J Roentgenol 2002, 179, 1245-1252, doi:10.2214/ajr.179.5.1791245

5. Page 8, line 328. Any data about some ongoing clinical trials related to this topic? 

Author Response

Response to Reviewer 1 Comments

We are grateful to Reviewer 1 for the critical comments and useful suggestions that have helped us improve our paper. As indicated in the responses that follow, we have considered all these comments and suggestions when revising our manuscript.

Point 1: Abstract. You can add a short sentences with the idea from the page 8, line 329: "no studies have demonstrated a suppressive effect on airway remodeling in childhood severe asthma".

Response 1: Thank you so much for your suggestion. We added the description as “No studies have demonstrated a suppressive effect on airway remodeling in childhood severe asthma, and further clinical trials using airway imaging analysis are needed to ascertain the inhibitory effect of biological drugs on airway remodeling in severe childhood asthma.” in the abstract (Page 1, Line 23-25).

Point 2: Page 2, line 54. Can you define the concept of the "spontaneous remission rate in childhood asthma?".

Response 2: Thank you very much for your important suggestion. The concept of spontaneous remission in the clinical study of Reference 11 was a 'complete remission' with normal pulmonary function and no bronchial hyperreactivity, without asthma symptoms or inhaled corticosteroid use. Therefore, we added a description of complete remission in the text(page 2, lines 61-64).

Point 3: Page 3, line 123. Packs year instead of " packs/year".

Response 3: Thank you for your suggestion. We corrected the description in the text as you suggested (Page 3, Line 131).

Point 4: Page 8, line 309. "Although CT airway analysis has been used mainly in adult studies to date [119,129,131,132]"- reference 132 is related to children : Marchac, V.; Emond, S.; Mamou-Mani, T.; Le Bihan-Benjamin, C.; Le Bourgeois, M.; De Blic, J.; Scheinmann, P.; Brunelle, F. Thoracic CT in pediatric patients with difficult-to-treat asthma. AJR Am J Roentgenol 2002, 179, 1245-1252, doi:10.2214/ajr.179.5.1791245.

Response 4: Thank you for your suggestion. As you pointed out, Reference 132 is a study of children with severe asthma, and we apologize for the wrong place to add it. In this pediatric clinical study, a 2D-CT airway analysis was performed rather than a 3D-CT airway analysis. Until recently, there have been no reports of clinical studies using 3D-CT airway analysis in children with asthma. We have revised the description to make the above discussion easier to understand (Page 8, Line 316-319).

Point 5: Page 8, line 328. Any data about some ongoing clinical trials related to this topic?

Response 5: Thank you very much for your kind and important suggestion. There was one clinical trial of prospective CT airway analysis in pediatric patients with severe asthma (ClinicalTrials.gov number, NCT05140889). We added the description of this ongoing clinical trial in the text (Page 8, Line 341-343).

Reviewer 2 Report

The research topic undertaken by the authors testifies to the excellent knowledge of asthma-related issues.

The abstract is correct, although I have noticed that the authors have forgotten the conclusions. They should be added here to make the abstract more informative.

The introduction is good. I have no objections here. However, I believe that since this is a review article - the authors should indicate what the selection of literature was, what databases they searched, and what keywords they used. On the other hand, this is not essential; on the other hand, it allows the reader to find reassurance that no essential articles have been accidentally missed.

In line 85 of FEV1, the abbreviation is not explained.

Figure 1 is attractive; I would like to know how the lines in the chart were estimated. Did the authors rely on any specific data and use it to create curves, or is this a proposition unsupported by numerical data?
I don't quite understand the blocks containing the perinatal, childhood, and adolescence/adult risk factors located just below the chart's axes. I am thinking about the location of these blocks in Figure 1. After all,  childhood is not suitable for people aged 10-30. Now, this is how you could read it on a chart. I was also wondering about the colors of these blocks; should they mean something? Do the blue color in the chart match the blue block below the chart? Probably not. So I suggest avoiding similar colors as it can be a bit confusing. Is the graph averaged for both genders?

Description of the action of biological drugs in treating asthma exacerbations and the schemes (Fig. 2 and 3) proposed by the authors - I have no criticism here. I believe that these issues have been discussed exhaustively.

I consider the content and graphic layout of the work to be correct—a proper selection of literature.

After adding the details I mentioned, I believe the article can be further considered and approved for publication.

Author Response

Response to Reviewer 2 Comments

We appreciate Reviewer 2 for taking the time to offer us the critical comments and useful suggestions that have helped us improve our paper. As indicated in the responses that follow, we have considered all these comments and suggestions when revising our manuscript.

Point 1: The abstract is correct, although I have noticed that the authors have forgotten the conclusions. They should be added here to make the abstract more informative.

Response 1: Thank you for your suggestion. We added the description as the conclusions in the abstract (Page 1, Line 23-25).

Point 2: The introduction is good. I have no objections here. However, I believe that since this is a review article - the authors should indicate what the selection of literature was, what databases they searched, and what keywords they used. On the other hand, this is not essential; on the other hand, it allows the reader to find reassurance that no essential articles have been accidentally missed.

Response 2: Thank you for your suggestion. We added the description of the selection of literature in the introduction (Page 2, Line 51-56).

Point 3: In line 85 of FEV1, the abbreviation is not explained.

Response 3: Thanks for the suggestion. We added the full name of FEV1 to the text (page 2, line 93).

Point 4: Figure 1 is attractive; I would like to know how the lines in the chart were estimated. Did the authors rely on any specific data and use it to create curves, or is this a proposition unsupported by numerical data? I don't quite understand the blocks containing the perinatal, childhood, and adolescence/adult risk factors located just below the chart's axes. I am thinking about the location of these blocks in Figure 1. After all, childhood is not suitable for people aged 10-30. Now, this is how you could read it on a chart. I was also wondering about the colors of these blocks; should they mean something? Do the blue color in the chart match the blue block below the chart? Probably not. So I suggest avoiding similar colors as it can be a bit confusing. Is the graph averaged for both genders?

Response 4: Thank you so much for your suggestion. The lines in the chart in Figure 1 are based on data from a longitudinal clinical study of lung function from childhood to adulthood in Reference #45. (45. McGeachie, M.J.; Yates, K.P.; Zhou, X.; Guo, F.; Sternberg, A.L.; Van Natta, M.L.; Wise, R.A.; Szefler, S.J.; Sharma, S.; Kho, A.T.; et al. Patterns of Growth and Decline in Lung Function in Persistent Childhood Asthma. N Engl J Med 2016, 374, 1842-1852, doi:10.1056/NEJMoa1513737.). We added the reference number to the Figure legend to help the reader's understanding (Page 3, Line 99).

As you pointed out, each block was not age-appropriate. We also agree that block colors and chart colors don't match and can be confusing. We added a dotted line to help the reader understand which age range corresponds to each block below the axis of the graph. We also changed the block color to black and white, and adjusted the graph colors to avoid similar colors (Page 3, Figure 1).